# Decision-Making Deficits in Elderly Can Be Alleviated by Attention Training

**DOI:** 10.3390/jcm8081131

**Published:** 2019-07-30

**Authors:** Marlen Schmicker, Inga Menze, David Koch, Ulrike Rumpf, Patrick Müller, Lasse Pelzer, Notger G. Müller

**Affiliations:** 1Neuroprotection Lab, German Center for Neurodegenerative Diseases (DZNE), Magdeburg 39120, Germany; 2Institute of Psychology, Otto von Guericke University, Magdeburg 39106, Germany; 3Medical Faculty, Otto von Guericke University, Magdeburg 39120, Germany; 4Center for Behavioral Brain Sciences, Magdeburg 39106, Germany

**Keywords:** visual working memory, selective attention, decision-making, aging, cognitive training, distractor inhibition, prominent deck B phenomenon

## Abstract

Decision-making is an important everyday function that deteriorates during normal aging. Here, we asked whether value-based decision-making can be improved in the elderly by cognitive training. We compared the effects of two training regimens on the performance in the Iowa Gambling Task (IGT), a real-life decision-making simulation task. Elderly participants (age 62–75 years) were randomized into three matched groups. The filter training (FT) group performed a selective attention task and the memory training (MT) group performed a memory storage task on five consecutive days. The control group (CG) did not perform another task besides the IGT. Only the FT group showed an improvement in IGT performance over the five days—the overall gain rose and the prominent deck B phenomenon decreased. The latter refers to the selection of cards associated with high gains and rare losses, which are nevertheless a disadvantageous choice as the frequent losses lead to a negative net outcome. As the deck B phenomenon has been associated with impaired cognitive abilities in aging, the positive effect of FT here is of special importance. In sum, attention training seems superior in improving decision-making in the elderly.

## 1. Introduction

In daily life, we have to make decisions all the time; they involve everyday matters (what to eat or what to wear) and larger issues (what kind of work to do, with whom to live). These decisions define an individual’s identity [1] and require various cognitive resources to assess advantages and disadvantages, risks and long-term gains of the options at hand. Aging is associated with a general cognitive decline and a decline in efficient and advantageous decision-making [2]. This is especially devastating as important decisions have to be made by seniors, too, e.g., regarding financial matters, medical care or retirement options. This underlines the need for improvement of decision-making in an aging population. 

It is, however, not trivial to assess decision-making abilities in a laboratory setting. In recent years, one of the most widely used instruments for investigating value-based decision-making under experimental conditions has become the Iowa Gambling Task (IGT). This task was originally developed by Bechara et al. [3] to detect prefrontal lobe damage, but nowadays is often applied in healthy subjects as well. Subjects are instructed to maximize their long-term gain and minimize their loss in an experimental context. They are presented with four decks of cards, whereby every card provides a monetary gain and a frequent loss. After drawing a card, the amount of money gained and the penalty are presented on a screen. Unbeknownst to the participants, decks A and B involve high gains but also substantial losses, resulting in a net loss when participants draw frequently from these decks (Figure 1). 

Deck C and D provide smaller direct gains, yet the losses per ten drawn cards are smaller than the gains, resulting in an overall long-term profit. In addition, the decks differ in their payoff schemes. Decks A and C have frequent losses, whereas decks B and D have infrequent losses [3]. The purpose of the task is to determine whether participants, through a trial-and-error process, are able to develop a preference for the advantageous decisions.

In general, healthy people should shift their deck selection from bad (A/B) to good choices (C/D) within 100 trials of the IGT [3]. However, some studies have observed that rather than developing a preference for both of the advantageous decks, participants tended to develop a preference for one advantageous deck (D) and one disadvantageous deck (B). This is explained by the fact that in both decks losses occur rarely, making these decks tempting, although only deck D leads to a long-term net gain. In particular, populations whose cognitive abilities have started to decline (e.g., older adults) are more likely to favor deck B [2,4,5]. This preference has been labeled the “prominent deck B phenomenon” [6] and has been related to limited cognitive resources [7].

The aim of improving decision-making in the elderly requires to identify the cognitive (sub)function that is most crucial for this complex capability. A number of cognitive functions have been proposed to be involved in decision-making, e.g., declarative memory [8], working memory (WM) [9], executive function and intelligence [10]. In order to assess which of these cognitive functions is essential for IGT performance, studies often use a dual-task, where the load of the cognitive function is manipulated; then the effects of this load manipulation on IGT performance is measured. In this way, it has been shown that increasing WM load decreased IGT capability [11,12]. Hence, it was postulated that WM is related to decision-making [4,12]. However, high attentional load has been shown to affect IGT performance as well. Subjects who performed a number monitoring task while completing the IGT were impaired in their decision-making [7]. They selected deck B more often because they were less aware of the payoff amount and instead concentrated on the low frequency of losses in deck B. Subjects who did not have to perform a secondary task and could devote their full attention to the IGT were able to take into account both frequency of losses and payoff amount; they were able to weigh previous gains and adjusted their choices accordingly. Unrestrained attention, therefore, seems to be important to avoid the prominent deck B phenomenon.

The latter observation suggests that attention might be especially important for IGT performance. In the past, it was shown that selective attention, i.e., the ability to filter out irrelevant information, correlates with memory capacity [13,14] and that training of filtering distractors can induce transfer effects that improve memory performance [15]. This led us to speculate that filter training (FT) may also enhance other cognitive functions. We demonstrated in an earlier study that FT over five days led to more advantageous decisions of young healthy subjects in the IGT than learning to store information in WM (memory training, MT) [16]. Unfortunately, in that study, only the total gain was measured, so training effects on the deck B phenomenon could not be assessed. In the present study, we compared the effects of both training regimens (FT and MT) on decision-making in elderly with a special focus on the deck B phenomenon. Knowing that (a) aging is associated with less efficient decision-making and (b) selective attention is crucial for IGT performance, we expected that FT should have a larger effect on IGT performance in the elderly than MT or no training (control group, CG), especially with respect to the Deck B phenomenon. 

## 2. Materials and Methods

### 2.1. Participants

Thirty-six healthy participants aged between 60 and 75 years were recruited through newspaper advertisements and database search at the German Centre for Neurodegenerative Diseases in Magdeburg, Germany. They had normal or corrected-to-normal vision, were right-handed, had no history of traumatic brain injury, stroke or psychiatric illness and were not taking psychotropic medication. Furthermore, they were screened using the Montreal Cognitive Assessment (MoCA) to make sure that all participants were cognitively unimpaired (inclusion criteria: minimum score of 26). Subjects were paid for their participation and, in addition, received an amount of money that was calculated as the mean gain of five IGT sessions (ranging from 8.25 € to 38.05 €; mean: 23.65 €). 

The study protocol was approved by the ethics committee of the University of Magdeburg (Germany) and all participants gave written informed consent in accordance with the Declaration of Helsinki. The study was registered as a clinical trial at clinicaltrial.gov (NCT03228446).

### 2.2. Procedure

The subjects were pseudo-randomly allocated to three groups for a single-blinded study design. Participants were not informed about the existence of different groups. One group received a filter training (FT), the second group underwent a memory training (MT) and the third did not perform any additional task (CG). One MT subject was excluded because he was familiar with the IGT before entering the study, another MT subject and three CG subjects were excluded due to low cognitive performance (MoCA under 26). In the end, we analyzed 31 data sets (12 FT, 10 MT, 9 CG; 69.2 years ± 3.6, 15 females; MoCA: 26-30 points). All groups performed the IGT on five consecutive sessions.

On the first day (Friday), all participants completed a cognitive test battery, including the initial screening using MoCA. Furthermore, they completed a digit span task and three subtests of the Test of Attentional Performance (TAP [17]): a two-back task, a divided attention task and a visual scanning task.

The active training groups (FT & MT) received 45 minutes of cognitive training daily, from Monday to Friday. FT and MT subjects also performed the decision-making IGT for five minutes on each day after training. For practical reasons, CG subjects performed the five IGT sessions on one day (Monday of the next week).

### 2.3. Stimuli and Tasks

#### 2.3.1. Training Tasks

We used an adapted version of a paradigm we had used in our earlier studies [15,16]. While participants of the FT had to indicate whether two simultaneously presented arrays of bars matched or not, MT subjects performed a computerized delayed match-to-sample task where two displays of bars were presented in succession. In FT targets were embedded within irrelevant distractors (bars in a different color, see Figure 2).

A red or green cue instructed subjects in FT to compare either only the red or green bars of the double array while ignoring bars of the other color. The arrays consisted of 4–6 relevant bars and the same number of irrelevant distractors. Participants were instructed to decide whether the simultaneously presented arrays matched in terms of the orientation of the relevant bars. This condition did not demand any memory storage. In MT after a neutral cue, the two arrays were presented consecutively with a delay of 500 ms at the center of the screen. The task was to decide whether one of the bars in the second array had changed its orientation relative to the first array. Hence, MT lacked the necessity to filter out irrelevant distractors.

MT and FT subjects pressed one of two buttons to respond. In half of the trials, the orientation of one target changed, and in the other half of the trials, no orientation change occurred.

The experiment ran under Matlab (R2013a, MathWorks, Inc., Natick, Massachusetts, United States) and the Psychtoolbox Version 3.0.12 and was presented on a laptop with a 15-inch screen and a resolution of 1024 × 1768 pixels. The display contained rectangular bars (0.28° × 0.72°) within a region covering 4° × 9.3° on a grey background (RGB = 189, 189, 189). In the filter training, stimuli arrays that had to be compared were placed 1.79° to the left and to the right of the central fixation cross.

Based on the finding that individual differences predict cognitive training benefits [18], we individually adjusted the difficulty of the training. Each subject’s performance in the last 10 trials was recorded and according to a fixed threshold (i.e., 80% correct) the difficulty of the following trials was adjusted. For example, if a subject achieved more than 80% correct responses in the first 10 trials with 4 items, the next 10 trials would contain 5 items; if performance was below 80%, then the next 10 trials would contain 4 items again. One training session (FT/MT) lasted 45 min. The number of total trials depended on an individual’s speed (faster-responding participants got more trials), but the session was automatically terminated after 400 trials. To assess training success, we compared the subject´s performance in the filter or memory task on the fifth day with that of the first day.

#### 2.3.2. The Iowa Gambling Task (IGT)

Decision-making was assessed with the Iowa Gambling Task (IGT) task, where participants gain or lose money by developing advantageous or disadvantageous decision strategies [3]. Each session consisted of 100 deck selections and we analyzed the selection behavior for each deck separately. Based on the scoring method reported by Stocco et al. [19], the payoff sensitivity (net outcome) was measured as the number of good deck draws (C + D) minus the number of bad deck draws (A + B) whereas the frequency sensitivity was defined as the difference between low loss frequency decks and high loss frequency decks: (B + D) − (A + C). The group average for selected decks over 5 training sessions and between session 5 and 1 as well as the prominent Deck B phenomenon (the difference between deck D selections and deck B selections [6]) were compared over all three groups.

In order to motivate our participants, a real amount of money was paid in €. We converted the fictive $-gain to real money by averaging the gain across the five sessions, then dividing it by 100 and converting it 1:1 to Euros (e.g., a fictive gain of 2000 $ was converted to 20 €).

Furthermore, instead of applying a standard pre-post design (i.e., IGT performance before and after training), we had our subjects perform the IGT on each of the five training days. This would allow us to quantify the amount of training necessary to improve decision-making.

### 2.4. Data Analysis

An a priori power analysis was performed using G × Power) [20] to estimate the sample size being necessary to detect training effects in IGT performance. Effect sizes for differences between the first and last day of training are specified by Cohens’s d, with d ≥ 0.2 indicating a small effect, d ≥ 0.5 a moderate effect, and d ≥ 0.8 a large effect [21]. Preliminary analyses from a previous study [15] indicated a large effect size (d = 0.89) of FT to improve performance on IGT in young adults. An estimated *n* of 16 participants were therefore needed to detect the effect as statistically significant with 95% power with one-tailed statistical tests. For repeated-measures ANOVA, an assumed large effect size f = 0.60 with the between factor ‘group’ (MT, FT, CG) and five measurements (day 1–5), the estimated sample size was *n* = 30 (10 participants per group).

For data analysis, age, education and performance in the cognitive test battery were compared between the three groups. Second, the improvements within the trained tasks for FT and MT were analyzed. Furthermore, group differences in the mean IGT gain, the payoff sensitivity, the frequency sensitivity, the deck selections over all days, the difference between selections of the first and the last day as well as the deck B phenomenon were analyzed.

For all analyses, two-way ANOVAs and two-way repeated measures ANOVAs were conducted. For repeated measures, group (FT, MT, CG) was used as the between-subjects factor, while session (1–5) represented the within-subjects factor. We calculated post-hoc t-tests using a Bonferroni correction for multiple comparisons and effect sizes for all gain and deck differences. The significance level adopted for all analyses was α = 0.05.

## 3. Results

### 3.1. Cognitive Baseline Assessment

In order to assure that there were no group differences in attention and working memory capacities, all participants underwent a cognitive test battery (TAP [17]) prior to the experiment. The groups did not differ in this test battery, nor in age and education levels (Table 1).

### 3.2. Training Effects in Memory and Filter Task

All subjects achieved the most difficult training levels of the adaptive attention and memory tasks; respectively, their overall performance in both tasks increased from day 1 to day 5. The FT group completed 93.8% attention trials of the highest difficulty level (6 targets and 6 distractors, 1st day: 75.5%) whereas the MT group achieved 59.4% memory trials with the highest load (6 targets) on the last day (1st day: 36.72%). In these most difficult conditions, FT subjects gave 69.4% (standard deviation = SD: 9.8%) and MT subjects 75.9% (SD: 5.3%) correct answers.

### 3.3. Training Effects on IGT Overall Gain

FT and MT subjects performed the IGT after each training session, while the CG group repeated the task five times on one day without any further task. Figure 3 displays the overall net gain of the 5 sessions. A two-way repeated measures ANOVA showed a significant main effect of the factor session (*F*(3.08,86.36) = 8.009, *p* < 0.001, *η*^2^ = 0.222), indicating that subjects gained more money in the later sessions. The session by group interaction was significant (*F*(6.17,86.36) = 4.844, *p* < 0.001, *η*^2^ = 0.257) indicating that the groups differed in their training success. Post-hoc t-tests revealed that FT led to a significantly higher money gain compared to MT and CG on the last training session.

Pairwise comparisons showed a significant increase of money gain from the first session to the last session in FT (*t*(11) = 6.71, *p* < 0.001, *d* = 2.56), whereas the differences for MT and CG did not become significant after Bonferroni correction (MT: *t*(9) = 1.91, *p* = 0.088, *d* = 0.51; CG: *t*(8) = 2.03, *p* = 0.077, *d* = 0.98).

On average, FT subjects gained 28.17 € over all five days (SD: 8.52 €; range: 11.59€–38.05€): the MT group earned 20.15 € (SD: 5.78 €; range: 8:25 €–30.00 €) and the CG subjects were paid 21.52 € (SD: 5.32 €, range: 15.35 €–29.55 €). Differences between training session one and five, *t*-values, *p*-values, effect sizes and corresponding between-group statistics are depicted in Table 2 and Table 3.

### 3.4. Training Effects on Payoff Sensitivity and Frequency Sensitivity

The net gain alone does not reveal whether subjects learned to prefer both, decks with low penalties (C/D) and decks with infrequent losses (B/D). Payoff sensitivity and frequency sensitivity [19] were calculated for each IGT session and a two-way repeated measures ANOVA was conducted. For payoff sensitivity, a significant main effect of session (*F*(4,112) = 6.436, *p* < 0.001, *η*^2^ = 0.187) and a significant interaction effect of session × group (*F*(8,112) = 4.308, *p* < 0.001, *η*^2^ = 0.235) was found. No significant effects for frequency sensitivity were found. Sensitivity values are displayed in Figure 4.

### 3.5. Training Effects on the Prominent Deck B Phenomenon

In order to assess subjects’ strategies in even more detail, the number of draws from each card deck was considered separately (Figure 5A–C). We found main effects for the factor session for decks A, B and D (deck A: *F*(2.85,79.84) = 4.562, *p* < 0.01, *η*^2^ = 0.140; deck B: *F*(2.73,76.32) = 3.009, *p* < 0.05, *η*^2^ = 0.097; deck D: *F*(2.70,75.37) = 4.935, *p* < 0.005, *η*^2^ = 0.150). Effects for session × group interactions were observed for card selections from deck A, B and D (deck A: *F*(5.70,79.84) = 2.680, *p* < 0.05, *η*^2^ = 0.161; deck B: *F*(5.45,76.32) = 3.092, *p* < 0.05, *η*^2^ = 0.181; deck D: *F*(5.38,75.37) = 3.913, *p* < 0.005, *η*^2^ = 0.218). For deck C no main or interaction effect was observed (Figure 5A).

One-sample *t*-tests and effect sizes revealed that the FT group learned to choose more cards from deck D and fewer cards from deck A and B over the training week (*p* < 0.01). There was a significant increase in deck C draws for the MT group and a significant decrease for deck A selection in the CG (*p* < 0.05) from the first to the last session. However, these changes did not survive the Bonferroni correction threshold. Detailed deck selection behavior for each group and differences between groups can be found as mean, *t*- and *p*- values and effects sizes in the Appendix A (Table A1, Table A2, Table A3, Table A4, Table A5, Table A6, Table A7 and Table A8).

In order to investigate the deck B phenomenon in particular, we calculated a two-way repeated measures ANOVA for the difference between deck D and deck B draws (Figure 5B). A statistically significant main effect of session was observed (*F*(2.72,76.13) = 4.506, *p* < 0.01, *η*^2^ = 0.139) and a significant interaction session × group (*F*(5.44,76.13) = 3.856, *p* < 0.005, *η*^2^ = 0.216). The FT group increased deck D and decreased deck B draws (*p* < 0.001, *d* > 4.921) whereas MT and CG groups did not show any significant change. Group comparisons revealed that D–B selections differed between FT and MT in session 3, 4 and 5; and between FT and CG in session 4 and 5. All statistical values for D–B can be found in the appendix (Table A9 and Table A10).

Moreover, the preference for advantageous decks after FT was confirmed by subtracting all deck selections from deck A, B, C and D in session 1 from all deck selections in session 5 for all groups (Figure 5C). The deck selection behavior changed most in the FT group and not nearly as much in the other groups. As expected FT subjects developed a preference for deck C and D (advantageous) and avoided draws from deck A and B (disadvantageous). Significant differences over the training week occurred for deck A, B and D in the FT group only (*p* < 0.01).

## 4. General Discussion

In the present study, only participants who received the FT improved their performance both in the trained filter task and in the IGT (net gain). Participants who underwent MT improved only in the trained memory task but did not show transfer effects on the IGT. Finally, participants of the control group, who did not receive a memory or filter training, did not improve over repeated versions of the IGT either. The interpretation of these finding is limited. Only small groups were compared, especially with high within-group variance.

### 4.1. Transfer Mechanism on IGT Performance

A closer look reveals that FT subjects learned to choose advantageous cards, i.e., they selected more cards from deck C & D and avoided decks A & B. In other words, they were more sensitive for the payoff scheme and the frequency of losses. Even with a small sample size, statistical significance is easily reached, because observed effect sizes (Cohen´s *d* and partial Eta square) are large. This implies a sufficient power to detect a significant training effect on the IGT performance. Training effects in FT started to emerge after the third session and became strongest on day 5. This indicates that there might have been room for further improvement with even more training sessions. The fact that neither the MT nor the CG group showed improvements in IGT performance renders the possibility unlikely that training of the IGT itself (through its repeated administration over five sessions) was responsible for the improvements in the FT group.

The role of selective attention in decision-making has so far rarely been a subject of research. The observed results confirm our previous training effects on IGT performance in young subjects [16] and provide further explanations for underlying mechanisms and relationships. In the IGT, participants are required to integrate rewards and losses, to remember the consequences of their decisions and to use them in subsequent decisions [22]. They have to focus on relevant outcomes and ignore high gains. But the ability to ignore irrelevant information is impaired in elderly persons [23,24]. This deficit may explain the preference for deck B in older adults and why attention training had a positive effect on IGT in this study: FT subjects might have learned to deal with additional, cognitive load and were able to weigh both the payoff frequency (one out of ten loss instead of 5:5 loss) and the amount of probabilistic loss ($250 instead of $1250) while selecting cards. Although the frequency sensitivity in FT did not change during the training week, subjects avoided to draw from deck A with the high frequent losses. The FT group was also more likely to choose deck D over B while MT and CG subjects used to select approximately equally often from deck B and D until the end of the training. The results indicate that MT and CG subjects were either still tempted to choose card decks with infrequent losses despite the amount of loss [25] or they did not develop a conscious strategy for choosing cards at all. The observed results could have been influenced by individual differences in response speed that are commonly observed among older healthy participants. Instead, reaction times in the neuropsychological assessment did not reflect systematic speed differences between groups. Another explanation assumes that FT boosted control mechanism for sensory markers. FT may guide attention towards less (emotionally) salient stimuli, while suppressing strong emotions induced by salient bottom-up stimuli [26]. By transferring their training-enhanced selection ability to filter out cognitive irrelevant markers, FT subjects may have learned to avoid disadvantageous selections (for a detailed discussion see [17]). Filter training may be highly beneficial for older subjects as it enhances remaining cognitive functions and is generalized to more everyday-related abilities. If a five-day filter training is effective enough should be investigated in further studies. All in all, our observations underline the importance of selective attention in decision-making.

### 4.2. Why Is Decision-Making Not Enhanced by Memory Training?

At first sight, the observation that MT subjects did not start to outperform the CG subjects over the consecutive training sessions is somewhat surprising. Indeed it could have been expected that memory training would affect the cognitive processes involved in remembering infrequent losses and high gains. However, it is known that older adults in spite of their intact memory are impaired in the IGT [22]. This again underlines that processes other than memory are crucial for effective decision-making in aging. Another explanation why MT subjects did not show transfer effects on decision-making might be the fact that the training was too exhausting and subjects simply had no resources left to improve in the IGT. It is known from dual-task experiments that increasing load of the WM task impairs IGT performance. Hence, performing a demanding WM task before the IGT might also have a rather detrimental influence. Against this speaks that increasing attentional load likewise reduces IGT performance in dual-task conditions [7]. So performing a filter task beforehand should have the same influence on the following IGT experiment. Furthermore, based on the training performance on the last day, the FT task was similarly demanding as the MT task. The explanation that MT but not FT depleted cognitive resources necessary for decision-making in the later IGT task is therefore unlikely.

In an earlier study, we were able to show that a similar attention training program also improved WM performance [15]. This suggests that filtering is a cognitive core function necessary in numerous complex tasks [18]. Training of filtering, therefore, has the potential to induce transfer effects on many other cognitive functions. In contrast, memory training seems to promote strategies by which decisions are only partially evaluated (amount or frequency of losses) and these factors are not weighed against each other (see also [15]).

### 4.3. Conclusions

Training to filter out irrelevant information improves the performance of elderly subjects in a decision-making task by alleviating disadvantageous behavior, namely the deck B phenomenon. We propose that a selective attention training can increase the ability to focus on all relevant factors of a task, in our case “loss frequency” and “probability for this loss”; which then leads to more advantageous choices (i.e., selecting deck D instead of deck B) and a higher net gain. In order to understand these mechanisms better, neuroimaging studies are necessary and possible far transfer effects of filter training on higher cognitive processes, such as problem-solving, should be investigated in more detail. Showing that decision-making is highly related to selective attentional filtering processes provides new advances as decisions have direct and immediate impact on economical, educational, medical and individual perspectives, especially at older ages. Given the ever-rising number of elderly people and the substantial consequences of disadvantageous decisions in late life, this approach bridges the interests of cognitive decline prevention and the theoretical investigation of underlying mechanisms. Therefore, clinicians should recommend training selective attention in addition to memory functions in order to prevent cognitive decline, especially disadvantageous decision-making. Mobile applications for cognitive training should also focus more on this ability. Currently, we are developing dedicated attention training programs, for example in the form of apps or using virtual reality based on real environments [27], that clinicians will be able to prescribe their patients.

## Figures and Tables

**Figure 1 jcm-08-01131-f001:**
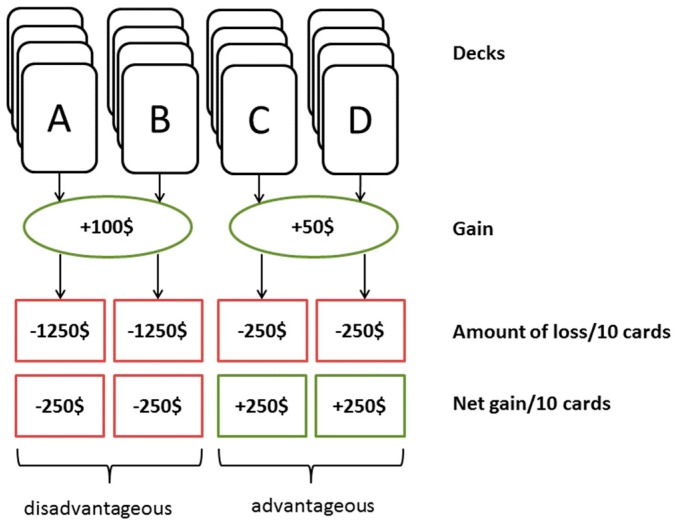
Payoff scheme for the Iowa Gambling Task. Drawing frequently from deck A or deck B results in net loss of −250 $ (disadvantageous decks), drawing from deck C and D results in a net gain of +250 $ (advantageous decks) [3].

**Figure 2 jcm-08-01131-f002:**
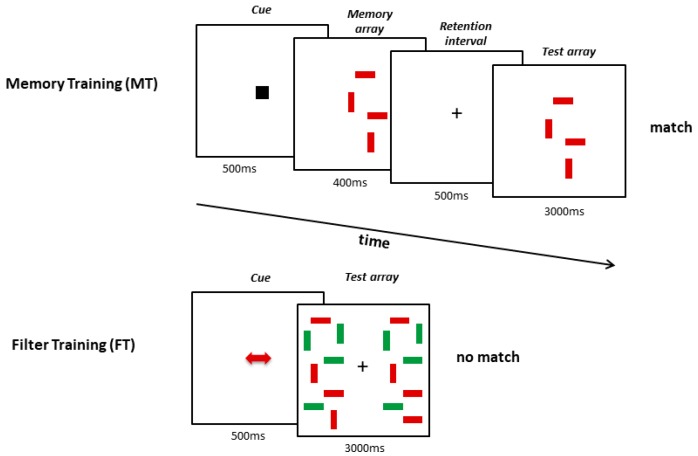
Schematic representation of Memory and Filter Training.

**Figure 3 jcm-08-01131-f003:**
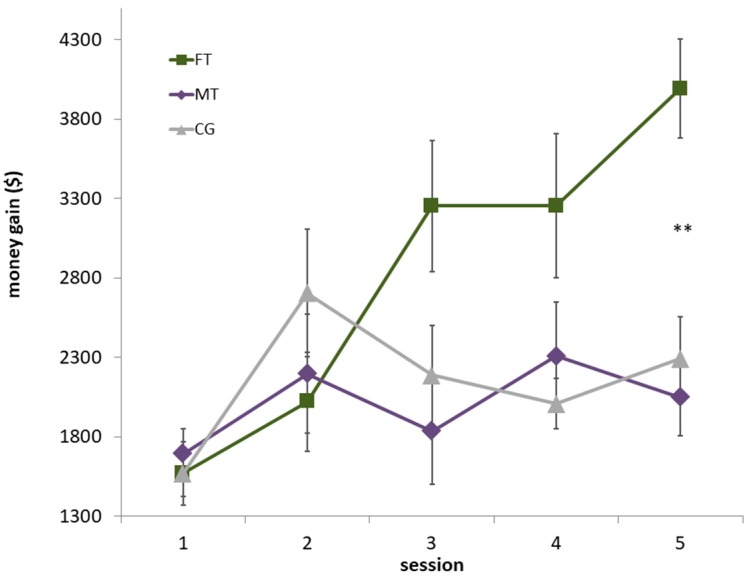
Influence of Filter Training (FT), Memory Training (MT) on IGT outcome compared to a passive Control Group (CG) displayed as means and standard errors; ** *p* < 0.001 for post-hoc comparison (Bonferroni corrected).

**Figure 4 jcm-08-01131-f004:**
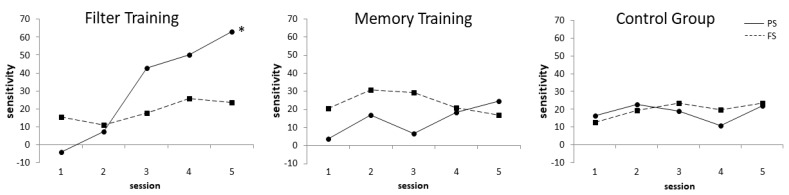
Payoff (PS) and Frequency (FS) Sensitivity. PS = (C + D) – (A + B); FS = (B + D) – (A + C), * *p* < 0.001 for day 1 vs. day 5 comparison.

**Figure 5 jcm-08-01131-f005:**
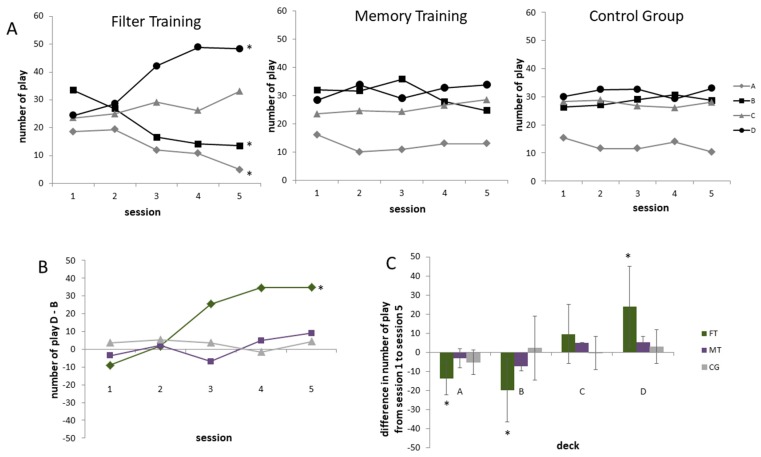
Deck selection over five sessions for FT, MT and CG. (**A**) The number of plays for the card decks A, B, C and D over five sessions displayed separately for FT, MT and CG. (**B**) The calculated Deck B phenomenon (deck D minus deck B) and (**C**) the difference between selected cards from deck A, B, C and D in session 1 and session 5 for all groups. * *p* < 0.01 after Bonferroni correction.

**Table 1 jcm-08-01131-t001:** Group comparisons for independent working memory and attention tasks.

		FT		MT		CG		ANOVA	
		Mean	SD	Mean	SD	Mean	SD	*F*	*p*
	age	69.60	3.12	67.70	4.32	70.20	3.20	1.33	0.281
	education	10.67	0.98	10.80	1.03	10.44	1.33	0.25	0.783
**WM**	n-back	11.90	1.97	11.90	1.97	11.11	2.57	1.53	0.859
	digit span forward	8.00	1.61	7.30	0.95	7.89	1.36	0.79	0.463
	digit span backward	6.45	2.16	6.10	1.52	6.89	2.03	0.40	0.677
**ATT**	divided attention auditory	15.91	0.30	15.80	0.42	15.56	1.33	0.51	0.605
	divided attention visual	15.09	2.88	15.30	1.49	14.89	1.54	0.90	0.915
	visual scanning	42.00	5.33	39.20	8.20	40.22	11.94	0.28	0.757

FT: *n* = 12, MT: *n* = 10, CG: *n* = 9. SD = standard deviation.

**Table 2 jcm-08-01131-t002:** Statistics for one-sample *t*-tests within groups (the last session compared to the first session) for overall IGT gains.

	FT	MT	CG
Δ session 5–1 (€)	24.18	3.55	7.22
*t*	6.71	1.91	2.03
*p*	**<0.001 ****	0.095	0.083
*d*	**2.56**	0.51	0.98

For all differences, the *t*-value, significance (*p*) and effect sizes (Cohen´s *d*) are displayed. Effect sizes higher than 1 and significant *p*-values are written in bold (** significant after Bonferroni correction).

**Table 3 jcm-08-01131-t003:** Statistics for group comparisons for 5 sessions for overall IGT gains.

	FT–MT (€)	*t*	*p*	*d*	FT–CG (€)	*t*	*p*	*d*
session 1	−1.21	−0.47	0.647	0.19	0.01	0.01	0.996	0.01
session 2	−1.74	−0.36	0.722	0.15	−6.84	−1.37	0.188	0.60
session 3	14.17	2.59	**0.017**	**1.11**	10.63	1.93	0.068	0.85
session 4	9.46	1.62	0.122	0.69	12.45	2.30	**0.033**	**1.01**
session 5	19.42	4.79	**<0.001 ****	**2.05**	16.97	3.97	**<0.001 ****	**1.75**

For all differences, the *t*-value, significance (*p*) and effect sizes (Cohen´s *d*) are displayed. *p* < 0.05 and *d* > 0.80 are highlighted in bold (** significant after Bonferroni correction).

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
