# Peer review of "Decision-Making Deficits in Elderly Can Be Alleviated by Attention Training"

_jcm, 2019, doi:10.3390/jcm8081131_

Reviewer 1 Report

This is an interesting study showing that cognitive training for the normally ageing people can improve their decision making skills. Although the paper pertains to the important issue and it is generally well-written, several questions should be clarified.

The description of the IGT lacks clarity. In particular, one cannot understand why card B is disadvantageous, taking into account that it -- in ten drawings -- it gives +$1000 gain and -$250 loss (see Fig. 1). Definitely, such terms should produce the net gain of +$750. Comparing this description with the original paper by Bechara et al. (1994), as well as other Bechara's papers, suggests that card B should give the same loss as card A, i.e., -$1250, although with different frequency. In other words, cards A and B should give the same net loss of -$250 (the same magnitude, different frequency), whereas the cards C and D ship give the net gain of +$250 (again, the same magnitude, different frequency). The authors' description suggests another solution, which is a bit surprising because they refer to the original paper by Bechara et al. (1994). This issue is important end needs clarification.

The sample size is very small. 36 participants were divided into there groups of 12 people each. After exclusion of some participants, the groups consisted of 10 or 11 people. One can argue that the statistical power was sufficient since some between-group differences crossed the boundary of significance. However, comparing 10 people with 11 is rather risky, the more so that the population of elderly people is quite differentiated, with great amount of within-group variance (much greater than in earlier stages of development). At the very least, this issue should be addressed as one of the serious limitations of this study.

The authors report incredibly great size effects, measured with Cohen's d (Table 2 and 3). Cohen's d equal 6 or 9 (!) suggests that the groups differed by six or nine standard deviations. In most training studies size effects rarely cross the value of 1, usually varying from 0 to 1. Are these computations really valid?

English is generally very good but I wonder how the following sentences was meant to look like: 

After drawing a card, participants are presented with both, the amount of money gained or lost with the actual card 

Author Response

Thank you for reviewing our manuscript and these valuable comments. We appreciate your thoughts and we think your suggestions largely contributed to an improvement of our manuscript.

We revised figure 1 regarding your suggestions (Bechara et al., 1994; 2005).

We agree that the main limitation should be emphasized and added the disadvantage of the small sample size to the very beginning of our discussion.

We apologize for wrong effect sizes. Now, the values are correct (but still very high): “Pairwise comparisons showed a significant increase of money gain from the first session to the last session in FT (t(1,11) = 6.71, p < .001, d = -2,557) whereas the differences for MT and CG did not become significant after Bonferroni correction (MT: t(1,9) = 1.91, p = .088, d = 1,275; CG: t(1,8) = 2.03, p = .077, d = -0.978).”

The sentence was changed into: After drawing a card, the amount of money gained and the penalty were presented on a screen.

Reviewer 2 Report

J. Clinical Med.

Decision-making deficits in elderly can be alleviated by attention training.

Schmicker et al.

Review:

Thank you for allowing me to review this manuscript describing the testing of an experimental protocol to improve decision-making in an aging population, as assessed using the IOWA Gambling Task (IGT). It is overall a robust study with good methodology and suitable statistical analysis. Alongside a control group of cognitively normal aged people, two other groups received either a selective attention task (filter training or FT); or a memory storage task (memory training, or MT) each day for a total duration of 5 days. There were clear significant task derived monetary gains at session 5 in the filter-training group, with better discrimination of net loss, therefore better command of / control over disadvantageous choice making. The data suggest that older people receiving attentional training improved their sensitivity to the frequency of losses.

Reviewers Comments: Decision is to accept the manuscript with minor amendments.

1). The abstract should include that the participants received a defined duration of training (ie:5 consecutive days of training).

2). More data should be presented regarding the participants baseline of responding. I am familiar with the Mini mental state examination (MMSE), however in this study the Montreal cognitive assessment was used. It is good practice that the authors have included the inclusion criteria of a minimum score of 26, and description that 3 subjects were excluded due to low cognitive score. However it would be advantageous to include a range of the MoCa scores of the 36 participants undergoing  testing.

3). Continuing from the point above, the introduction could be expanded to speak a little about the prevalence of cognitive decline in the general population. For example that 2% of people in the 60-70 year age bracket have a dementia state (most usually Alzheimer’s Disease), and that neuropathological changes in the brain occur far in advance of florid symptoms. This being so, cognitive capacity can be a very variable construct.

4). In the discussion it should be considered how much filtering training is beneficial and for what reasons? Is 5 days a minimally effective exposure, would more training reap higher rewards?

5). Looking to the discussion and conclusions I would urge some attempt to include some suggestions for best clinical practice to support robust cognition in the elderly. Specifically how clinicians can support selective attentional filtering processes, for example in memory clinics, or even home-based structured therapies.

Author Response

Thank you for reviewing our manuscript and your great suggestions. We think your suggestions largely contributed to an improvement of our manuscript.

1)      We added the duration of the training to the abstract.

2)      Now, the MOCA range is mentioned.

3)      We included the decline of general cognitive functions in higher age.

4)      We added the following part: “FT may guide attention towards less (emotionally) salient stimuli, while suppressing strong emotions induced by salient bottom-up stimuli [26]. By transferring their training-enhanced selection ability to filter out cognitive irrelevant markers FT subjects may have learned to avoid disadvantageous selections (for a detailed discussion see [17]). Filter training may be highly beneficial for older subjects as it enhances remaining cognitive functions and is generalized to more everyday-related abilities. If a five-day filter training is effective enough should be investigated in further studies. All in all, our observations underline the importance of selective attention in decision-making.”

5)      We agree that practical recommendations are very important. Unfortunately, it is not easy to implement such training in everyday life or memory clinics without using technical support. We therefore added the following part to out conclusion: “Therefore, clinicians should recommend training selective attention in addition to memory functions in order to prevent cognitive decline, especially disadvantageous decision-making. Mobile applications for cognitive training should also focus more on this ability.  Currently, we are developing dedicated attention training programs, for example in the form of apps or using virtual reality based on real environments [27] , that clinicians will be able to prescribe their patients.” For more information, please see https://www.frontiersin.org/articles/10.3389/fpsyg.2019.01046/full

Round  2

Reviewer 1 Report

All my concerns have been properly addressed. However, the astonishingly big effect size figures are still present in the Tables, although they have been corrected in the main body of there paper. This discrepancy must be removed.

Author Response

Thank you for this suggestion. We revised the effect sizes in table 2 and 3 and adapted them in the manuscript text.